# CoLa: A Choice Leakage Attack Framework to Expose Privacy Risks in Subset Training

## Abstract

Subset training, where models are trained on a carefully chosen portion of data rather than the entire dataset, has become a standard tool for scaling modern machine learning. From coreset selection in vision to large-scale filtering in language models, these methods promise scalability without compromising utility. A common intuition is that training on fewer samples should also reduce privacy risks. In this paper, we challenge this assumption. We show that subset training is not privacy free: the very choices of which data are included or excluded can introduce new privacy surface and leak more sensitive information. Such information can be captured by adversaries either through side-channel metadata from the subset selection process or via the outputs of the target model. To systematically study this phenomenon, we propose CoLa (Choice Leakage Attack), a unified framework for analyzing privacy leakage in subset selection. In CoLa, depending on the adversary's knowledge of the side-channel information, we define two practical attack scenarios: Subset-aware Side-channel Attacks and Black-box Attacks. Under both scenarios, we investigate two privacy surfaces unique to subset training: (1) Training-membership MIA (TM-MIA), which concerns only the privacy of training data membership, and (2) Selection-participation MIA (SP-MIA), which concerns the privacy of all samples that participated in the subset selection process. Notably, SP-MIA enlarges the notion of membership from model training to the entire data–model supply chain. Experiments on vision and language models show that existing threat models underestimate the privacy risks of subset training: the enlarged privacy surface not only retains training membership leakage but also exposing selection membership, extending risks from individual models to the broader ML ecosystem.

## 1 Introduction

The scale of modern datasets has made training on the full corpus increasingly impractical. To address this, practitioners routinely employ subset training, where only a carefully chosen ratio of data is used. This paradigm is adopted not only for efficiency but also to improve data quality, since selection can remove redundancy and noise while retaining informative samples. Subset training spans diverse applications: coreset selection (Bachem et al., 2015; Munteanu et al., 2018; Mirzasoleiman et al., 2020) in vision, dataset pruning (Sorscher et al., 2022; Yang et al., 2022; Qin et al., 2023), active learning (Sener & Savarese, 2018; Ducoffe & Precioso, 2018; Agarwal et al., 2020) in general ML, and large-scale deduplication (Lee et al., 2022), filtering (Rae et al., 2021), and sampling (Gunasekar et al., 2023; Peng et al., 2025; Wettig et al., 2024) in language model pretraining.

While subset training is widely celebrated for these benefits, its privacy implications remain under-explored (Zhao & Zhang, 2025). A common intuition suggests that fewer training samples should imply less privacy leakage (Dong et al., 2022). Yet this reasoning overlooks an important fact: *the choices made during subset selection themselves encode signals about which data were included and which were excluded.* These signals can be inherited through shifts in the data distribution or model behavior, making them exploitable by adversaries.

We ask the fundamental question: *Does subset training actually reduce privacy leakage?* Our answer is *no*. We show that subset training introduces new attack surfaces: not only is the included data that used for training compromised, but the excluded data discarded from training can also become

vulnerable due to correlations introduced by the selection mechanism. In other words, due to the data-oriented nature of the subset selection process, beyond the training data leakage emphasized by traditional MIA (Shokri et al., 2017; Hu et al., 2022), the choice signals further extend privacy risks from individual models to the broader data–model supply chain. Accordingly, we define two complementary privacy surfaces: *Training-membership MIA (TM-MIA)*, which resembles traditional MIA by focusing on the membership of training data, and *Selection-participation MIA (SP-MIA)*, a privacy surface tailored to subset training that focuses on membership at the data selection level.

To systematically study membership leakage under these privacy surfaces, we propose **CoLa (Choice Leakage Attack)**, a framework that leverages choice signals in a principled way to conduct attacks across different surfaces. CoLa captures risks under two complementary settings: (i) a *Subset-aware Side-channel* setting, where the adversary has access to the target model's outputs and selection metadata (e.g., the selection algorithm and the inclusion ratio); and (ii) a *Black-box* setting, where the adversary observes only model outputs and is aware that subsetting may have been used, without knowing any selection metadata. Extensive results show that for both privacy surfaces under these two attack settings, CoLa can substantially strengthen the attack performance. In short, subset training does not guarantee privacy; it enlarges the attack surface of modern ML pipelines and highlights the need to protect privacy across the entire data–model supply chain. We summarize our contributions as follows:

- We provide the first systematic definition and exploration of the membership leakage problem under subset training. This novel attack scenario reveals a severe privacy risk in the subset selection process: not only is the privacy of training data compromised, but the data excluded during selection is also at risk.

- We propose CoLa (Choice Leakage Attack), a framework tailored to subset selection that leverages choice signals in a principled way for more reliable membership inference, while seamlessly unifying diverse attack settings and surfaces.

- Experiments across both vision and language models confirm the broad capability of CoLa. For example, in the black-box setting, the AUC of CoLa on Pythia-160M surpasses 80% under SP-MIA, where all baseline methods fail.

## 2 RELATED WORKS

**Subset training and data-efficient learning.** A large body of research has explored how to reduce the cost of large-scale training by operating on subsets of data. Coreset selection constructs small but representative subsets that approximate training on the full data (Bachem et al., 2015; Munteanu et al., 2018; Mirzasoleiman et al., 2020; Yang et al., 2024b). Dataset pruning removes redundant or low-value samples to improve efficiency and generalization (Sorscher et al., 2022; Yang et al., 2022; Qin et al., 2023; Maharana et al., 2023; Tan et al., 2024). Active learning queries the most informative examples to reduce annotation cost (Sener & Savarese, 2018; Ducoffe & Precioso, 2018; Agarwal et al., 2020; Borsos et al., 2020; Margatina et al., 2021). In large-scale language models, deduplication and filtering pipelines are routinely applied to eliminate noise and improve training quality (Lee et al., 2022; Rae et al., 2021; Raffel et al., 2023; Gao et al., 2020a). These techniques have been extensively studied for efficiency and utility, but their privacy consequences remain largely underexplored.

**Membership inference attacks.** Membership inference attacks (MIAs) are among the most widely studied privacy threats in machine learning. Early work by Shokri et al. (2017) proposed shadow models to train attack classifiers distinguishing members from nonmembers. Subsequent methods exploited confidence scores, loss values, or gradients (Yeom et al., 2018; Sablayrolles et al., 2019; Carlini et al., 2022b). MIAs have been demonstrated in supervised learning, federated learning, and large language models (Nasr et al., 2018; Hu et al., 2022; Li et al., 2025), motivating defenses such as differential privacy (Abadi et al., 2016) and adversarial regularization (Nasr et al., 2018). This body of work reveals how models trained on fixed datasets can memorize and leak sensitive information. However, they primarily focus on constructing membership signals in a one-shot manner, with these signals being tightly coupled to a specific model. We find such model-oriented signal less effective in the context of subset training. Leveraging the unique characteristics of the subset selection process, we instead construct membership signals in a data-oriented manner.

**Synthetic data and privacy.** Synthetic data generation has been studied as a way to train models without exposing raw datasets, with the promise of stronger privacy (Hu et al., 2024; Tan et al., 2025). However, subsequent research has shown that synthetic datasets can still leak sensitive information about the original data, including membership and attributes (Stadler et al., 2022; van Breugel et al., 2023; Zhao & Zhang, 2025). Rather than analyzing risks inherent in *synthetic data generation pipelines*, we turn to *subset training with real data*, where high-fidelity samples remain but the selection process itself exposes a distinct and overlooked channel of privacy leakage.

# 3 PROBLEM SETTING

## 3.1 MEMBERSHIP INFERENCE UNDER SUBSET TRAINING

Let $D_0 \subseteq \mathcal{X} \times \mathcal{Y}$ denote the original dataset that undergoes a subset selection procedure. A selector $\text{Sel}(\cdot; r)$ with a given selection ratio $r$ partitions $D_0$ into two disjoint sets: the **included data** $I$ used for training, and the **excluded data** $E$ that are discarded:

$$(I, E) = \text{Sel}(D_0; r), \text{ with } I \cap E = \varnothing, \ I \cup E = D_0, |I|/|D_0| = r. \tag{1}$$

Following the standard MIA pipeline (Shokri et al., 2017), we further denote by $O$ the **outside data** that never enter the selection process. A model $f_\theta$ is trained solely on $I$. This partition naturally induces two types of membership inference task:

**Training-membership MIA (TM-MIA).** This attack takes the model itself as the attack surface and membership is defined solely by the training data. A sample $x$ is a member if $x \in I$ and nonmember if $x \in E \cup O$. This forms a natural and widely adopted threat model, as the model is the most direct output of the ML system. This setting is consistent with conventional MIAs (Shokri et al., 2017; Carlini et al., 2022b).

**Selection-participation MIA (SP-MIA).** However, when the attack surface is enlarged to the entire data–model pipeline, membership expands from only the training data to a much larger portion

Figure 1: Privacy surfaces under subset training.

of all collected data. As shown in Figure 1, we refer to the collected data as selection members, where a sample $x$ is a member if $x \in I \cup E$ and a non-member if $x \in O$. Its membership cannot be explained by direct model memorization, but instead reveals *choice leakage*, a side-channel signal from the subset selection process of the data-model supply chain. Such choice leakage risk is severe as it exposes a system's selection preferences. Once the data–model supply chain is exposed to privacy risks, the entire pipeline, from raw data to model outputs, becomes vulnerable to malicious manipulation. **To our knowledge, this is the first work to systematically investigate this perspective.**

Both tasks can be framed as binary hypothesis tests over a scoring function $s : \mathcal{D}_0 \to \mathbb{R}$, which measures the likelihood of a sample $x$ belonging to the respective member set. Given $\mathcal{D}_0 = I \cup E \cup O$, the member–nonmember partitions are:

$$\mathcal{M}_{\text{TM}} = I, \qquad \mathcal{N}_{\text{TM}} = E \cup O, \tag{2}$$

$$\mathcal{M}_{\text{SP}} = I \cup E, \ \ \mathcal{N}_{\text{SP}} = O. \tag{3}$$

The goal is to design a scoring function $s(x)$ that distinguishes $\mathcal{M}$ from $\mathcal{N}$ under both definitions.

## 3.2 ADVERSARY KNOWLEDGE

Subset training changes not only the definition of membership but also the adversary's potential knowledge and capabilities. We consider two complementary scenarios:

**Subset-aware side-channel attacks.** In line with the common assumption in prior MIAs, the adversary can query the deployed model $f_\theta$ and observe its outputs (e.g., prediction labels or confidence scores). In addition, it has access to *side information about the selection process*, such as the strategy used (e.g., coreset selection, pruning, filtering) or the approximate inclusion ratio. Such an assumption is realistic: pruning papers routinely report retained percentages to justify efficiency–utility

trade-offs, active learning and coreset methods describe selection strategies for reproducibility, and large-scale LLM pipelines release dataset cards documenting filtering heuristics, inclusion ratios, or deduplication statistics (Cohen-Addad et al., 2021; Biderman et al., 2023a; Dubey et al., 2024; Yang et al., 2024a). Crucially, this information reflects only high-level rules, not the exact membership of individual samples. Our attack targets precisely this gap: even when only the selection algorithm or ratio is public, an adversary can exploit this side-channel to infer which specific samples were included or excluded, thereby exposing *choice leakage* in subset training.

**Black-box attacks.** Here the adversary can only query the deployed model $f_\theta$ and observe its outputs. The entire subset selection stage is hidden, so the adversary must rely solely on the observable behavior of the trained model or the intrinsic data-specific information. This setting captures the most restrictive and widely assumed threat model in prior MIA research (Hu et al., 2022).

## 4 METHOD

### 4.1 CHALLENGES OF MEMBERSHIP INFERENCE UNDER SUBSET TRAINING

In conventional MIA, success comes from exploiting overfitting: models tend to assign systematically higher confidence to their training data than to non-members. Under subset training, however, this signal becomes entangled. Figure 2 illustrates this using the LiRA attack signal from (Carlini et al., 2022b) on a model trained on $I$ selected from $D_0$ by Glister (Killamsetty et al., 2021b). The dataset used here is CIFAR10 and the model is ResNet18. Since the selector is designed to make training on $I$ approximate the effect of training on $I \cup E$, the confidence distributions of included, excluded, and outside samples exhibit more complex overlaps: **(i)** $I$ concentrates at high confidence, $E$ shifts lower, while outside data often show a bimodal distribution; **(ii)** in TM-MIA, $I$ and $E \cup O$ remain partly separated but overlap substantially at high confidence; **(iii)** in SP-MIA, the distribution of $I \cup E$ largely overlaps with that of outside data, making the groups difficult to distinguish. This overlap complexity shows that model-oriented signals are no longer sufficient under subset training, highlighting the need for data-oriented alternatives.

### 4.2 CHOICE LEAKAGE ATTACK

**Motivation.** Just as models can overfit to their training data, subset selectors can *overfit at the selection level*: by design they preferentially reselect examples that match their implicit criteria (e.g., high informativeness, low noise, or strong representativeness). This persistent re-selection introduces a stable bias in the choice process that itself serves as a reliable membership signal. We exploit this *inclusion stability*, the tendency of a sample to be repeatedly chosen across multiple trials, as the core signal for our attack.

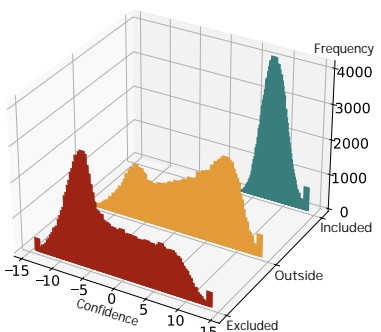

Figure 2: Signal distributions of three groups of data under subset training.

Specifically, we approximate many different candidate combinations by constructing a series of overlapping subsets ("windows") $\{W_i \subseteq D_0\}_{i=1}^m$, where $m$ is the number of windows, to capture inclusion-stable samples. Each $W_i$ represents one plausible candidate set the selector might face; by examining the selector's decisions on a sample across these windows, we reveal whether it is consistently favored.

**Subset-aware side-channel attack.** In the side-channel setting, the adversary knows both the selector $\mathrm{Sel}(\cdot; r)$ and the selection ratio $r \in (0, 1]$. For each window $W_i$, we run $\mathrm{Sel}(\cdot; r)$ and record whether $x \in W_i$ is selected by the selector, and get its evidence $e(x, W_i)$ in the current window:

$$e(x, W_i) = \mathbb{1}[x \in \mathrm{Sel}(W_i; r)]. \tag{4}$$

Suppose in the window construction, $x$ appears in $n$ out of $m$ windows; by aggregating the selection evidence across these windows, we obtain its *inclusion count*:

$$t(x) = \sum_{i=1}^{n} e(x, W_i), \tag{5}$$

Table 1: Results for vision models under the subset-aware side-channel attack setting. Results are averaged over 9 coreset selection methods. *Intensity* denotes the selection ratio $r$ (Light: $r = 0.2$, Medium: $r = 0.4$, Heavy: $r = 0.6$, Extensive: $r = 0.8$). Best results per row are in bold.

| Intensity | Setting | NN | | NN_top3 | | NN_cls | | LiRA | | CoLa | |
|---|---|---|---|---|---|---|---|---|---|---|---|
| | | AUC | TPR@5%FPR | AUC | TPR@5%FPR | AUC | TPR@5%FPR | AUC | TPR@5%FPR | AUC | TPR@5%FPR |
| Light | SP-MIA | 51.23 ±2.56 | 5.83 ±1.34 | 51.77 ±3.02 | 3.34 ±4.45 | 51.59 ±2.66 | 6.37 ±2.22 | 51.26 ±4.85 | 6.43 ±3.70 | **61.39** ±2.48 | **14.24** ±2.02 |
| | TM-MIA | 64.00 ±12.15 | 12.13 ±5.96 | 61.57 ±15.05 | 8.93 ±13.52 | 67.24 ±16.18 | 19.30 ±17.14 | 69.86 ±22.08 | 15.74 ±18.19 | **83.77** ±2.44 | **42.19** ±4.51 |
| Medium | SP-MIA | 52.33 ±3.56 | 6.31 ±1.48 | 53.59 ±4.30 | 3.56 ±2.28 | 54.51 ±4.51 | 6.64 ±1.58 | 54.99 ±4.69 | 5.31 ±0.42 | **81.93** ±3.50 | **42.66** ±5.81 |
| | TM-MIA | 59.84 ±12.84 | 10.80 ±4.97 | 60.37 ±10.53 | 2.91 ±3.32 | 66.84 ±11.79 | 12.51 ±5.85 | 62.96 ±13.69 | 4.61 ±2.52 | **88.53** ±2.55 | **60.10** ±7.62 |
| Heavy | SP-MIA | 52.21 ±3.83 | 12.20 ±15.48 | 53.20 ±4.85 | 2.80 ±2.43 | 53.53 ±5.94 | 12.37 ±15.44 | 53.69 ±5.68 | 4.26 ±1.77 | **96.86** ±2.60 | **88.60** ±5.51 |
| | TM-MIA | 55.00 ±9.59 | 19.31 ±26.18 | 52.40 ±10.78 | 1.67 ±2.04 | 57.44 ±11.06 | 19.63 ±26.72 | 52.61 ±11.77 | 2.81 ±2.32 | **89.06** ±1.90 | **60.36** ±5.87 |
| Extensive | SP-MIA | 55.64 ±5.31 | 7.59 ±1.68 | 59.56 ±6.39 | 4.00 ±2.92 | 56.66 ±5.90 | 7.60 ±1.87 | 61.54 ±8.36 | 5.09 ±2.68 | **92.20** ±6.94 | **91.86** ±7.23 |
| | TM-MIA | 61.41 ±6.63 | 10.99 ±2.61 | 60.13 ±12.20 | 4.21 ±4.15 | 62.80 ±7.52 | 11.27 ±2.73 | 59.66 ±12.03 | 4.63 ±3.77 | **80.74** ±8.23 | **49.76** ±6.98 |

where $t(x)$ is the number of times $x$ is selected, For fair comparison, the windows are constructed as sliding windows with fixed intervals and cyclic wrapping (details are provided in Section 5), thus each data appears in exactly the same number of windows. Hence, the exposure count $n$ is constant across all $x$ and serves only as a scaling factor in our score function. This also highlights the motivation behind our multi-shot membership signal: rather than relying on a single output, choice leakage signal is derived from *how consistently a sample is selected across different selections.* The membership score $s_{\text{Side}}(x)$ is obtained by aggregating evidence across windows:

$$s_{\text{side}}(x, n, r) = w\big(t(x); n, r\big), \qquad (6)$$

where $w$ is a monotone weighting function. From a statistical perspective, if each inclusion is a Bernoulli trial, then $t(x) \sim \text{Binomial}(n(x), p(x))$ where $p(x)$ is the probability of a data to be included. Given the selection ratio $r$, the expected inclusion count under random choice is $r \cdot n(x)$. We can therefore design $w$ as a smooth monotone mapping centered around $r \cdot n(x)$:

$$w\big(t(x); n(x), r\big) \;=\; \frac{\sigma\big(\kappa(t(x) - r \cdot n(x)))\big)}{Z(n(x), r)}, \quad \sigma(u) = \frac{1}{1 + e^{-u}}, \; \kappa > 0, \qquad (7)$$

where $\kappa$ controls the slope and $Z$ is a normalization constant (depending only on $n(x), r$) that does not affect relative ranking. Since the ratio $r \in (0, 1]$ and each sample has the same exposure count $n$. Without loss of generality, we therefore adopt the following simplified scoring function:

$$w(t(x); n) \;=\; \sigma\big(t(x) - \tfrac{n}{2}\big) = \frac{1}{1 + e^{-(t(x) - \frac{n}{2})}}. \qquad (8)$$

This formulation monotonically amplifies scores of samples with high inclusion counts and constrains the range by $n$, which makes scores comparable across windows. Finally, under both TM-MIA and SP-MIA, the decision is made by thresholding:

$$\hat{y}(x) = \mathbb{1}[s_{\text{side}}(x) \geq \tau], \qquad (9)$$

where $\tau$ is a decision threshold. Samples that are more stably selected as included data across windows will receive higher scores and are thus more likely to be classified as training members.

**Black-box attack.** In this setting, the subset selection process remains a black box to the adversary, and no direct selection metadata is available. Guided by our general motivation of *inclusion stability* (samples that are repeatedly reselected across plausible candidate sets reveal membership), we infer stable inclusion by identifying samples that consistently act as geometric representatives across windows. Specifically, for each window we perform unsupervised embedding clustering to locate representative samples. Formally, let $f(\cdot)$ be an embedding model. For each window $W_i \subseteq \mathcal{D}_I$, we compute embeddings $f(x), x \in W_i$, and perform k-means clustering (Ahmed et al., 2020) in the embedding space. Each sample $x \in W_i$ is then assigned to a cluster $c(x; W_i)$, and we measure its distance to the corresponding cluster centroid $d(x, W_i) = \|f(x) - c(x; W_i)\|_2$. The distance is used to serve as the evidence:

$$e(x, W_i) = \mathbb{1}\big[d(x, W_i) \leq Q_{0.5}(W_i)\big], \qquad (10)$$

where $Q_{0.5}(\cdot)$ is the median distance among all samples in $W_i$. The formal definitions of the inclusion count and exposure count follow the same formulation as in Eq. 5, with the only difference that the evidence $e(x, W_i)$ is redefined as Eq. 10 under the current black-box setting.

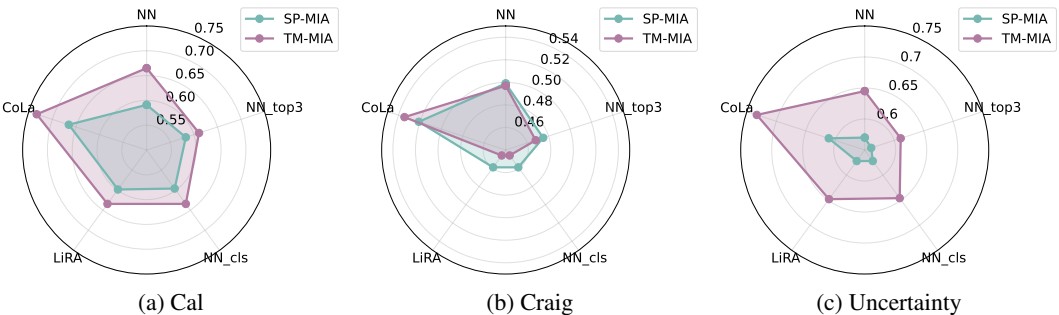

| (a) Cal | (b) Craig | (c) Uncertainty |

Figure 3: The MIA performance on vision models under black-box setting.

Here, to capture multi-shot stability, since the evidence for each data now related the distance to its centroid in each window $W_i$, we apply a weighted score function which reveals not only the inclusion count but also the actual distance it receives:

$$s_{\text{black}}(x) = w(t(x); n)/\bar{d}(x), \tag{11}$$

where $\bar{d}(x) = \frac{1}{t(x)} \sum_{i: x \in W_i} d(x, W_i)$ denotes the average clustering distance of sample $x$ across the windows in which it is included. This design ensures that samples consistently close to centroids across many windows receive higher scores. The weighting function $w(t; n)$ follows the same formulation as in Eq. 8. Finally, similar to the side-channel setting, membership is determined by thresholding:

$$\hat{y}(x) = \mathbb{1}[s_{\text{black}}(x) \geq \tau]. \tag{12}$$

This unsupervised formulation enables membership inference even without any knowledge of the underlying subset selection metadata. The inclusion stability-based pipeline of CoLa naturally unifies different attack surfaces within a single framework, thereby facilitating coordinated attacks.

## 5 EXPERIMENTS

### 5.1 SETUPS

**Models and Datasets.** We conduct experiments on both vision and language models. For the vision side, without loss of generality, we use ResNet-18 trained on CIFAR-10. We evaluate the performance on both subset-aware side-channel attacks and black-box attacks. For language models, since training multiple LMs from scratch is computationally expensive, we restrict our study to black-box attacks. Leveraging the rich open-source models in NLP and following the setup in (Meeus et al., 2024), we use deduplicated models from the Pythia (Biderman et al., 2023b) and GPT-Neo (Black et al., 2021) families, specifically pythia-70m, pythia-160m, and gpt-neo-125m, all trained on the MIMIR dataset (Gao et al., 2020b; Duan et al., 2024). From the MIMIR dataset, we select two subsets, arXiv and PubMed Central, and evaluate each under two split settings: 'arxiv_ngram_1_0.8', 'arxiv_ngram_13_0.2', 'pubmed_central_ngram_13_0.8', and 'pubmed_central_ngram_13_0.2', where '13_0.8' denotes removing non-member examples that share $> 80\%$ 13-gram overlap with members.

In the black-box attacks for vision models, we derive embeddings from the activations just before the final linear layer of a shadow model that shares the target model's architecture. The shadow model is trained using the GradMatch method (Killamsetty et al., 2021a) (distinct from the MIA methods evaluated in our paper) with a selection rate of 0.5. For language models, due to the various lengths of each sequence, we obtain fixed-dimensional embeddings using a dedicated embedding model; by default we use 'all-MiniLM-L6-v2' (Reimers & Gurevych, 2019; Thakur et al., 2021).

For CoLa, the default interval is set to 500 for vision models and 100 for language models, with the window size to be 20,000 and 1,000, respectively. In black-box attacks, the number of clusters is fixed at 5. Ablation studies are provided in Section 5.4.

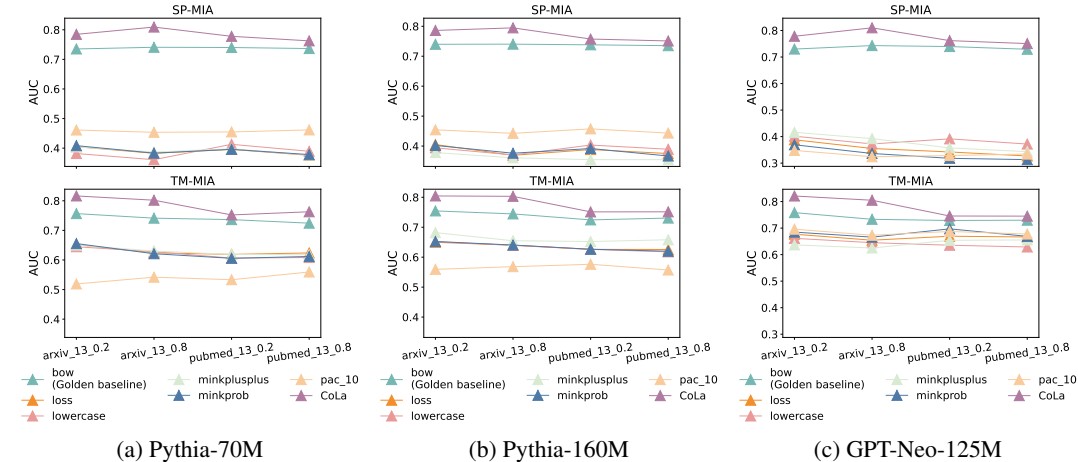

Figure 4: The MIA performance on language models under black-box setting.

**Subset Selection Methods.**    For vision models, we select nine representative dataset pruning methods from different categories. Specifically, we include decision boundary based methods such as DeepFool (Ducoffe & Precioso, 2018) and Contrastive Active Learning (Cal) (Margatina et al., 2021); the bi-level optimization based method Glister (Killamsetty et al., 2021b); error based methods including Forgetting (Toneva et al., 2018) and GraNd (Paul et al., 2021); the uncertainty based method Least Confidence (denoted as Uncertainty) (Coleman et al., 2020); the gradient matching based method Craig (Mirzasoleiman et al., 2020); and geometry based methods such as Contextual Diversity (Agarwal et al., 2020) and Herding (Welling, 2009). These methods cover a broad range of perspectives on dataset pruning, from boundary sensitivity to optimization criteria, error contribution, uncertainty, gradient alignment, and geometric diversity. The selection ratio is set to 0.2, 0.4, 0.6, and 0.8. For language models, as discussed in the previous subsection, we adopt a commonly used data filtering strategy that has been systematically studied in (Meeus et al., 2024; Duan et al., 2024), and consider two deduplication strengths, namely '13_0.8' and '13_0.2'.

**Baseline MIA Methods.**    For vision models, we consider four baselines: NN, NN_top3, and NN_Cls (Shokri et al., 2017; Salem et al., 2018), which use the model's output logits, the top-3 logits, and the combination of logits with class labels as membership signals, respectively, as well as LiRA (Carlini et al., 2022b), which fits Gaussian distributions and leverages the likelihood to infer membership. The shadow model used in each baseline method is set to 8. For language models, we consider six baselines, including the loss (Yeom et al., 2018), Lower (lowercase) (Carlini et al., 2021), Min-K% (minkprob) (Shi et al., 2023), Min-K%++ (minkplusplus) (Zhang et al., 2024), Pac (pac_10) (Ye et al., 2024), and the Golden baseline Bag_of_Words (bow) (Meeus et al., 2024). Here, bow serves as a performance reference: methods performing below it are regarded as ineffective.

**Evaluation Metrics.**    In most MIA studies (Hisamoto et al., 2020; Carlini et al., 2022a; Li et al., 2025), attack performance is typically evaluated by aggregating over all possible thresholds using the AUC score. We adopt the same practical evaluation metric in our experiments. We also report True Positive Rate at low False Positive Rate (TPR@Low FPR) (Carlini et al., 2022b), which is an important metric in MIAs and measures the detection rate at a meaningful threshold.

## 5.2 RESULTS UNDER SUBSET-AWARE SIDE-CHANNEL ATTACKS

A subset-aware side-channel attack is a type of attack specific to the subset selection process. Its success indicates that current practices of disclosing meta information about subset selection are unsafe and can lead to privacy leakage.

Table 1 reports the average MIA results across different coreset selection methods we consider for vision models (detailed results for each method are provided in Appendix A.2). As shown, in the relatively simple TM-MIA setting, baseline methods can still perform reasonably well, which is expected since this setting closely resembles traditional MIAs (Shokri et al., 2017; Hu et al., 2022) for which these baselines were originally designed.

However, in the SP-MIA setting that is unique to sub-set training, baseline methods largely fail (AUC close to 50%), indicating their inability to effectively distinguish between included and excluded data. Fundamentally, this stems from the fact that baseline methods rely heavily on model outputs; as illustrated in Figure 2, included and excluded data exhibit output distributions that are highly similar to other data, resulting in poor separability. However, this does not mean that privacy cannot be compromised under SP-MIA. In contrast, CoLa achieves strong performance in both TM-MIA and SP-MIA settings, thanks to its multi-shot, data-centric membership signal that tightly aligns with the subset selection process and captures fine-grained data interactions, thereby enabling better separability. Moreover, we observe that as

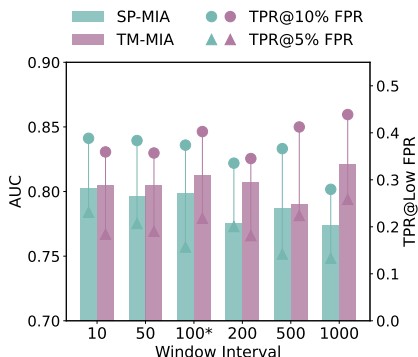

Figure 5: The influence of the window size on the MIA performance.

the selection ratio (*Intensity*) increases, the risk of privacy leakage becomes more severe, highlighting the significant vulnerability of the subset selection process as a potential side channel.

## 5.3 RESULTS UNDER BLACK-BOX ATTACKS

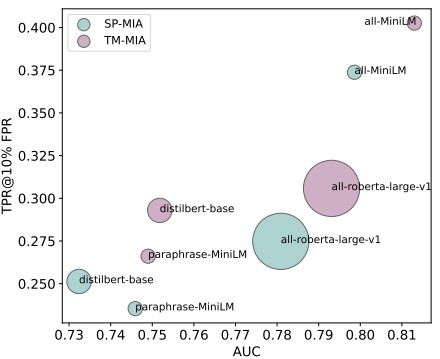

Figure 6: The influence of the embedding model on the MIA performance.

In the black-box attack setting, we study both vision and language models. Language model subset selection often relies on heuristic semantic filtering or deduplication, rather than the formally defined selection algorithms and ratios common in vision, which makes it naturally suited to black-box analysis. In this scenario, the adversary has no access to any meta information about the selection procedure. Consequently, a successful membership inference attack under these conditions indicates that the subset selection process itself—much like model training—can implicitly reveal private information about the data. This implies that privacy risks arising from subset selection must be addressed proactively: mitigating them requires careful design choices and safeguards before the selection process is executed.

The results for vision models and language models are shown in Figure 3 and Figure 4, respectively. For vision models, we adopt three representative selection methods: Cal (Margatina et al., 2021), Craig (Mirzasoleiman et al., 2020), and Uncertainty (Coleman et al., 2020). As illustrated in Figure 3, under the black-box setting, SP-MIA remains more challenging than TM-MIA. Moreover, CoLa consistently outperforms the baselines by about 5% in AUC across all experiments, demonstrating strong attack capability. For language models, this contrast is even more pronounced. As shown in Figure 4, all baseline methods except CoLa perform worse than the bow baseline, indicating that they essentially fail in the context of subset selection MIA. Furthermore, while SP-MIA and TM-MIA results are relatively close for CoLa, the baselines exhibit a sharp gap, with SP-MIA close to random guessing (AUC around 50%), and TM-MIA reaches only about 60%.

## 5.4 ABLATION STUDIES.

**Influence of Window Construction.** In Figure 5, we present an ablation study on the influence of window interval, conducted with Pythia-160m on the arxiv_ngram_13_0.8 dataset. Several observations can be made: first, regardless of the window interval size, the performance under SP-MIA is consistently lower than that under TM-MIA, highlighting its greater challenge. Second, the choice of window interval size does not substantially affect the performance of CoLa. In SP-MIA, increasing the size reduces the exposure count $n$ of each data sample, which makes the inclusion signal coarser and leads to a slight performance drop. However, this drop remains marginal.

**Influence of Embedding Model.** As a data-centric MIA method, CoLa achieves a clear decoupling from the target model. As discussed earlier, it derives the membership signal by reallocat-

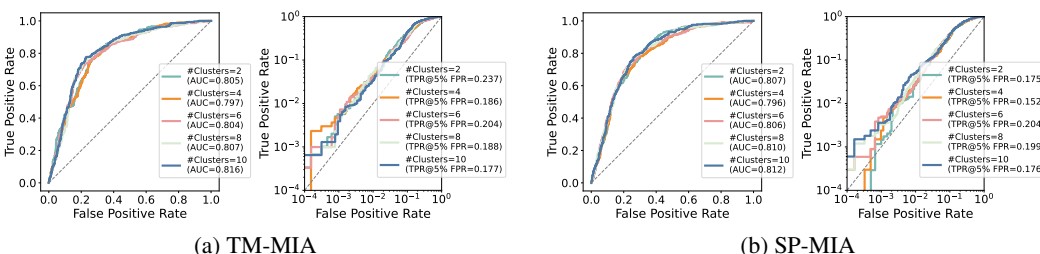

(a) TM-MIA                                          (b) SP-MIA

Figure 7: The MIA performance on language models under the black-box setting.

ing data combinations based on overfitting at the selection level. For language data, the inherent inconsistency in format and length requires the use of a dedicated embedding model in this reallocation process. To examine the effect of embedding model choice, we conduct an ablation study beyond the default all-MiniLM-L6-v2, considering three alternatives: paraphrase-MiniLM-L6-v2 (paraphrase-MiniLM), distilbert-base-nli-stsb-mean-tokens (distilbert-base), and all-roberta-large-v1. The results are shown in Figure 6, where the circle size indicates the parameter scale of each embedding model. We observe that

Table 2: Subset-aware Side-channel attacks under different vision models and datasets.

| Setting | ResNet18-CIFAR100 | | VGG19-CIFAR10 | | VGG19-CIFAR100 | |
|---|---|---|---|---|---|---|
| | AUC | TPR@5%FPR | AUC | TPR@5%FPR | AUC | TPR@5%FPR |
| SP-MIA | 67.28 ±1.36 | 19.05 ±1.17 | 64.98 ±2.03 | 17.15 ±2.12 | 70.31 ±1.64 | 21.23 ±1.81 |
| TM-MIA | 85.53 ±2.07 | 38.46 ±1.43 | 81.43 ±1.43 | 40.35 ±1.65 | 86.67 ±2.36 | 42.10 ±1.39 |

different embedding models have a noticeable impact on inference performance, particularly on TPR at low FPR. Moreover, larger model size does not necessarily translate into better performance, highlighting the importance of choosing an appropriate embedding model. Nevertheless, the results remain generally acceptable across all choices (with AUC consistently above 70% and TPR@10% FPR above 25%). How to customize embedding models for MIA under subset selection is a meaningful question, which we leave for future work.

**Results under Different Vision Models and Datasets.** In Table 2, we further conduct subset-aware side-channel attack on the CIFAR-100 dataset with the VGG19 model to verify whether CoLa remains reliable across different vision datasets and models. The selection ratio here is set to 0.2. As can be observed, CoLa consistently works well across various vision model–dataset combinations, revealing its general applicability. Specifically, attacks on VGG19 are more pronounced than on ResNet18 under the same setting, and CIFAR-100 is more vulnerable than CIFAR-10. Moreover, the observation that SP-MIA is more challenging than TM-MIA is consistent with previous findings.

**Influence of Clustering.** In Figure 7, we study the effect of varying the number of clusters used for embedding clustering in the black-box setting. Beyond the default choice of 5, we further consider values between 2 and 10 and report the corresponding AUC curves and TPR@5% FPR. The results show that, for both SP-MIA and TM-MIA, the clustering number has only a marginal effect on performance.

## 6  CONCLUSION

In this work, we take the first step toward systematically understanding the privacy risks of subset training. Contrary to the common intuition that training on fewer samples should reduce privacy leakage, we demonstrate that the very choices made during subset selection can themselves become exploitable signals, exposing both included and excluded data to membership inference. To capture this phenomenon, we introduced CoLa, a unified framework that leverages choice patterns to construct robust membership signals. Across both vision and language models, under both subset-aware side-channel and black-box settings, CoLa consistently outperforms existing baselines, revealing that subset training does not mitigate but instead amplifies privacy leakage. Our findings highlight that privacy risks extend beyond model outputs to the data–model supply chain itself. We hope this work motivates future efforts toward designing selection mechanisms and training pipelines that are not only efficient and scalable but also privacy-preserving.

## ETHICS STATEMENT

This work focuses on understanding privacy risks in subset training through systematic analysis of membership inference attacks (MIAs). Our study is purely methodological and does not involve human subjects or personally identifiable information. All datasets used are publicly available benchmark datasets (e.g., CIFAR, GSM8K, CodeAlpaca), and we complied with their intended use and licensing terms. We emphasize that the proposed Choice Leakage Attack (CoLa) is presented as a research contribution to highlight potential vulnerabilities in modern training pipelines, not to enable misuse. Our findings are intended to inform the community about inherent privacy risks and to guide the development of stronger defenses. No proprietary or sensitive data was used, and no deployed models were targeted in this study. In line with research integrity, we also note that Large Language Models (LLMs) were only employed for literature review support and polishing of textual presentation (e.g., improving fluency and figure/table captions). LLMs were not involved in technical design, experimental implementation, or data analysis.

## REPRODUCIBILITY STATEMENT

We have made every effort to ensure the reproducibility of our work. All datasets used in this paper are publicly available, and their sources are clearly cited in the main manuscript. The implementation details of our methods, including models used, attack configurations, and evaluation protocols, are described in Section 5.1. We also provide ablation studies and additional experiments in Section 5.4 to validate the generality of our findings. Upon acceptance, we will release the full source code, configuration files, and scripts for evaluation to facilitate verification and future research.

## LLM DISCLAIMER

LLMs were used only occasionally for language polishing, aiming to improve fluency and readability. All technical ideas, experimental designs, analyses, conclusions, writing were developed and carried out entirely by the authors. The authors have full responsibility for the final text.

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

# A APPENDIX

## A.1 THE PRIVACY THREATS BEHIND DATA-MODEL SUPPLY CHAIN

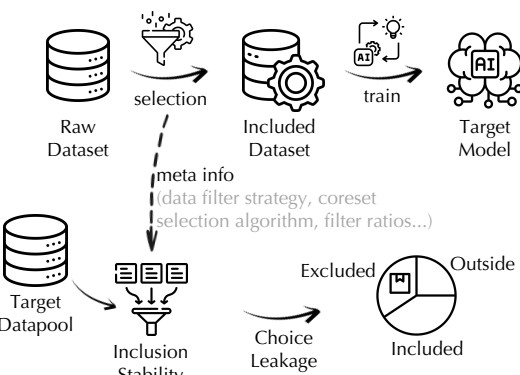

Figure 8: Choice Leakage Attack (CoLa) across the data–model supply chain. CoLa augments conventional MIA by exploiting subset selection metadata leaked along the data–model supply chain. By identifying which samples are more likely to pass selection, it not only strengthens membership inference but also enables adversaries to craft tailored threats.

As shown in Figure 8, the data–model supply chain describes the pipeline from raw data collection, through subset selection and model training, to the deployment of a target model. In this process, subset selection plays a central role: only a fraction of the raw dataset is included for training, while others are excluded or remain outside. The metadata of this selection process (e.g., filtering strategies, coreset algorithms, or filter ratios) introduces new privacy surfaces. Such information can inadvertently leak "choice signals" that reveal which samples are more likely to be included in training, thereby extending the privacy risk beyond conventional training data exposure.

CoLa (Choice Leakage Attack) directly exploits this vulnerability by leveraging selection metadata to strengthen membership inference. Unlike traditional MIAs that focus solely on the trained model's outputs, CoLa targets the entire supply chain, identifying which samples are predisposed to pass the selection process. Such choice leakage risk is severe as it not only amplifies the risk of inferring membership but also exposes a system's selection preferences. Once the data–model supply chain is exposed to privacy risks, the entire pipeline, from raw data to model outputs, becomes vulnerable to malicious manipulation. For example, adversaries may learn proxies of the selection rule and craft targeted poisoning or backdoor examples that are more likely to bypass filtering and enter training.

Table 3: The results of vision models under Subset-aware Side-channel attacks and the subset selection method used here is Cal (Margatina et al., 2021).

| Intensity | Setting | NN | | NN_top3 | | NN_cls | | LiRA | | CoLa | |
|---|---|---|---|---|---|---|---|---|---|---|---|
| | | AUC | TPR@5%FPR | AUC | TPR@5%FPR | AUC | TPR@5%FPR | AUC | TPR@5%FPR | AUC | TPR@5%FPR |
| Light | SP-MIA | 0.499 | 0.050 | 0.501 | 0.055 | 0.508 | 0.053 | 0.512 | 0.054 | 0.602 | 0.122 |
| | TM-MIA | 0.759 | 0.207 | 0.676 | 0.166 | 0.784 | 0.257 | 0.737 | 0.182 | 0.855 | 0.442 |
| Medium | SP-MIA | 0.553 | 0.072 | 0.573 | 0.056 | 0.582 | 0.074 | 0.587 | 0.058 | 0.789 | 0.372 |
| | TM-MIA | 0.763 | 0.165 | 0.759 | 0.097 | 0.812 | 0.227 | 0.784 | 0.092 | 0.878 | 0.620 |
| Heavy | SP-MIA | 0.589 | 0.077 | 0.603 | 0.000 | 0.630 | 0.087 | 0.624 | 0.054 | 0.963 | 0.856 |
| | TM-MIA | 0.729 | 0.123 | 0.721 | 0.000 | 0.772 | 0.172 | 0.736 | 0.058 | 0.895 | 0.642 |
| Extensive | SP-MIA | 0.634 | 0.091 | 0.637 | 0.000 | 0.647 | 0.092 | 0.651 | 0.036 | 0.957 | 0.954 |
| | TM-MIA | 0.717 | 0.116 | 0.707 | 0.061 | 0.736 | 0.128 | 0.690 | 0.026 | 0.849 | 0.573 |

Table 4: The results of vision models under Subset-aware Side-channel attacks and the subset selection method used here is Contextual Diverstiy (Agarwal et al., 2020).

| Intensity | Setting | NN | | NN_top3 | | NN_cls | | LiRA | | CoLa | |
|---|---|---|---|---|---|---|---|---|---|---|---|
| | | AUC | TPR@5%FPR | AUC | TPR@5%FPR | AUC | TPR@5%FPR | AUC | TPR@5%FPR | AUC | TPR@5%FPR |
| Light | SP-MIA | 0.540 | 0.067 | 0.539 | 0.053 | 0.548 | 0.073 | 0.544 | 0.052 | 0.633 | 0.118 |
| | TM-MIA | 0.706 | 0.125 | 0.716 | 0.070 | 0.755 | 0.161 | 0.756 | 0.072 | 0.798 | 0.347 |
| Medium | SP-MIA | 0.598 | 0.094 | 0.594 | 0.000 | 0.614 | 0.088 | 0.610 | 0.056 | 0.846 | 0.465 |
| | TM-MIA | 0.751 | 0.158 | 0.708 | 0.000 | 0.792 | 0.160 | 0.729 | 0.049 | 0.908 | 0.656 |
| Heavy | SP-MIA | 0.507 | 0.051 | 0.502 | 0.000 | 0.506 | 0.500 | 0.500 | 0.000 | 0.982 | 0.904 |
| | TM-MIA | 0.502 | 0.074 | 0.482 | 0.000 | 0.516 | 0.048 | 0.477 | 0.000 | 0.898 | 0.631 |
| Extensive | SP-MIA | 0.494 | 0.056 | 0.494 | 0.027 | 0.497 | 0.052 | 0.497 | 0.042 | 0.967 | 0.966 |
| | TM-MIA | 0.500 | 0.053 | 0.490 | 0.000 | 0.502 | 0.052 | 0.490 | 0.041 | 0.843 | 0.386 |

## A.2 RESULTS OF VISION MODELS UNDER DIFFERENT SUBSET SELECTION METHODS

In Table 1, we report the average results of vision models across nine subset selection methods. For clarity, Tables 3–11 present the results for each method separately, providing a more straightforward view of the attack performance.

Table 5: The results of vision models under Subset-aware Side-channel attacks and the subset selection method used here is Craig (Mirzasoleiman et al., 2020).

| Intensity | Setting | NN | | NN_top3 | | NN_cls | | LiRA | | CoLa | |
|---|---|---|---|---|---|---|---|---|---|---|---|
| | | AUC | TPR@5%FPR | AUC | TPR@5%FPR | AUC | TPR@5%FPR | AUC | TPR@5%FPR | AUC | TPR@5%FPR |
| Light | SP-MIA | 0.495 | 0.046 | 0.500 | 0.000 | 0.497 | 0.048 | 0.441 | 0.039 | 0.637 | 0.172 |
| | TM-MIA | 0.567 | 0.087 | 0.500 | 0.000 | 0.573 | 0.102 | 0.602 | 0.064 | 0.825 | 0.411 |
| Medium | SP-MIA | 0.513 | 0.066 | 0.588 | 0.054 | 0.580 | 0.086 | 0.598 | 0.055 | 0.819 | 0.367 |
| | TM-MIA | 0.595 | 0.137 | 0.693 | 0.043 | 0.717 | 0.134 | 0.716 | 0.045 | 0.858 | 0.518 |
| Heavy | SP-MIA | 0.575 | 0.076 | 0.614 | 0.052 | 0.628 | 0.082 | 0.629 | 0.051 | 0.969 | 0.876 |
| | TM-MIA | 0.624 | 0.114 | 0.628 | 0.030 | 0.701 | 0.122 | 0.647 | 0.034 | 0.888 | 0.562 |
| Extensive | SP-MIA | 0.624 | 0.092 | 0.655 | 0.000 | 0.653 | 0.096 | 0.664 | 0.000 | 0.960 | 0.959 |
| | TM-MIA | 0.674 | 0.110 | 0.666 | 0.000 | 0.700 | 0.119 | 0.623 | 0.000 | 0.842 | 0.545 |

Table 6: The results of vision models under Subset-aware Side-channel attacks and the subset selection method used here is DeepFool (Ducoffe & Precioso, 2018).

| Intensity | Setting | NN | | NN_top3 | | NN_cls | | LiRA | | CoLa | |
|---|---|---|---|---|---|---|---|---|---|---|---|
| | | AUC | TPR@5%FPR | AUC | TPR@5%FPR | AUC | TPR@5%FPR | AUC | TPR@5%FPR | AUC | TPR@5%FPR |
| Light | SP-MIA | 0.494 | 0.057 | 0.500 | 0.000 | 0.489 | 0.054 | 0.441 | 0.039 | 0.637 | 0.172 |
| | TM-MIA | 0.556 | 0.092 | 0.500 | 0.000 | 0.530 | 0.084 | 0.221 | 0.000 | 0.825 | 0.411 |
| Medium | SP-MIA | 0.494 | 0.051 | 0.501 | 0.048 | 0.492 | 0.050 | 0.500 | 0.050 | 0.845 | 0.480 |
| | TM-MIA | 0.649 | 0.088 | 0.550 | 0.000 | 0.642 | 0.088 | 0.397 | 0.011 | 0.926 | 0.700 |
| Heavy | SP-MIA | 0.496 | 0.053 | 0.507 | 0.000 | 0.496 | 0.052 | 0.509 | 0.042 | 0.979 | 0.900 |
| | TM-MIA | 0.494 | 0.053 | 0.429 | 0.016 | 0.484 | 0.054 | 0.424 | 0.000 | 0.902 | 0.643 |
| Extensive | SP-MIA | 0.526 | 0.096 | 0.643 | 0.062 | 0.545 | 0.097 | 0.645 | 0.067 | 0.956 | 0.954 |
| | TM-MIA | 0.592 | 0.142 | 0.571 | 0.037 | 0.602 | 0.140 | 0.574 | 0.038 | 0.858 | 0.572 |

Table 7: The results of vision models under Subset-aware Side-channel attacks and the subset selection method used here is Forgetting (Toneva et al., 2018).

| Intensity | Setting | NN | | NN_top3 | | NN_cls | | LiRA | | CoLa | |
|---|---|---|---|---|---|---|---|---|---|---|---|
| | | AUC | TPR@5%FPR | AUC | TPR@5%FPR | AUC | TPR@5%FPR | AUC | TPR@5%FPR | AUC | TPR@5%FPR |
| Light | SP-MIA | 0.500 | 0.053 | 0.500 | 0.000 | 0.514 | 0.059 | 0.530 | 0.051 | 0.618 | 0.141 |
| | TM-MIA | 0.572 | 0.099 | 0.500 | 0.000 | 0.706 | 0.176 | 0.741 | 0.071 | 0.854 | 0.475 |
| Medium | SP-MIA | 0.503 | 0.056 | 0.500 | 0.000 | 0.548 | 0.068 | 0.559 | 0.056 | 0.818 | 0.464 |
| | TM-MIA | 0.529 | 0.098 | 0.500 | 0.000 | 0.695 | 0.139 | 0.724 | 0.064 | 0.851 | 0.517 |
| Heavy | SP-MIA | 0.501 | 0.500 | 0.499 | 0.045 | 0.499 | 0.050 | 0.498 | 0.050 | 0.986 | 0.943 |
| | TM-MIA | 0.540 | 0.830 | 0.480 | 0.000 | 0.546 | 0.840 | 0.460 | 0.034 | 0.921 | 0.661 |
| Extensive | SP-MIA | 0.585 | 0.080 | 0.640 | 0.071 | 0.585 | 0.081 | 0.748 | 0.084 | 0.791 | 0.787 |
| | TM-MIA | 0.646 | 0.107 | 0.640 | 0.074 | 0.649 | 0.107 | 0.648 | 0.080 | 0.653 | 0.407 |

Table 8: The results of vision models under Subset-aware Side-channel attacks and the subset selection method used here is Glister (Killamsetty et al., 2021b).

| Intensity | Setting | NN | | NN_top3 | | NN_cls | | LiRA | | CoLa | |
|---|---|---|---|---|---|---|---|---|---|---|---|
| | | AUC | TPR@5%FPR | AUC | TPR@5%FPR | AUC | TPR@5%FPR | AUC | TPR@5%FPR | AUC | TPR@5%FPR |
| Light | SP-MIA | 0.495 | 0.048 | 0.500 | 0.000 | 0.492 | 0.045 | 0.545 | 0.062 | 0.608 | 0.135 |
| | TM-MIA | 0.477 | 0.033 | 0.500 | 0.000 | 0.422 | 0.000 | 0.883 | 0.129 | 0.829 | 0.384 |
| Medium | SP-MIA | 0.504 | 0.055 | 0.497 | 0.044 | 0.503 | 0.049 | 0.496 | 0.045 | 0.864 | 0.494 |
| | TM-MIA | 0.367 | 0.007 | 0.545 | 0.045 | 0.448 | 0.024 | 0.586 | 0.044 | 0.874 | 0.516 |
| Heavy | SP-MIA | 0.494 | 0.050 | 0.499 | 0.048 | 0.495 | 0.048 | 0.497 | 0.051 | 0.992 | 0.949 |
| | TM-MIA | 0.404 | 0.039 | 0.541 | 0.060 | 0.440 | 0.020 | 0.555 | 0.059 | 0.871 | 0.480 |
| Extensive | SP-MIA | 0.527 | 0.062 | 0.598 | 0.073 | 0.533 | 0.060 | 0.600 | 0.079 | 0.984 | 0.984 |
| | TM-MIA | 0.598 | 0.131 | 0.757 | 0.118 | 0.651 | 0.134 | 0.771 | 0.120 | 0.895 | 0.502 |

Table 9: The results of vision models under Subset-aware Side-channel attacks and the subset selection method used here is GraNd (Paul et al., 2021).

| Intensity | Setting | NN | | NN_top3 | | NN_cls | | LiRA | | CoLa | |
|---|---|---|---|---|---|---|---|---|---|---|---|
| | | AUC | TPR@5%FPR | AUC | TPR@5%FPR | AUC | TPR@5%FPR | AUC | TPR@5%FPR | AUC | TPR@5%FPR |
| Light | SP-MIA | 0.563 | 0.087 | 0.584 | 0.126 | 0.563 | 0.114 | 0.575 | 0.153 | 0.562 | 0.137 |
| | TM-MIA | 0.843 | 0.206 | 0.918 | 0.389 | 0.937 | 0.571 | 0.950 | 0.584 | 0.878 | 0.483 |
| Medium | SP-MIA | 0.498 | 0.048 | 0.498 | 0.047 | 0.497 | 0.050 | 0.499 | 0.052 | 0.754 | 0.344 |
| | TM-MIA | 0.535 | 0.103 | 0.471 | 0.019 | 0.573 | 0.104 | 0.471 | 0.018 | 0.902 | 0.680 |
| Heavy | SP-MIA | 0.493 | 0.047 | 0.500 | 0.051 | 0.493 | 0.047 | 0.501 | 0.050 | 0.909 | 0.774 |
| | TM-MIA | 0.557 | 0.119 | 0.387 | 0.011 | 0.562 | 0.118 | 0.384 | 0.012 | 0.859 | 0.606 |
| Extensive | SP-MIA | 0.505 | 0.054 | 0.502 | 0.047 | 0.506 | 0.054 | 0.503 | 0.048 | 0.839 | 0.826 |
| | TM-MIA | 0.572 | 0.110 | 0.378 | 0.005 | 0.556 | 0.109 | 0.380 | 0.019 | 0.712 | 0.498 |

Table 10: The results of vision models under Subset-aware Side-channel attacks and the subset selection method used here is Herding (Welling, 2009).

| Intensity | Setting | NN | | NN_top3 | | NN_cls | | LiRA | | CoLa | |
|---|---|---|---|---|---|---|---|---|---|---|---|
| | | AUC | TPR@5%FPR | AUC | TPR@5%FPR | AUC | TPR@5%FPR | AUC | TPR@5%FPR | AUC | TPR@5%FPR |
| Light | SP-MIA | 0.516 | 0.053 | 0.521 | 0.052 | 0.512 | 0.054 | 0.510 | 0.053 | 0.574 | 0.171 |
| | TM-MIA | 0.853 | 0.407 | 0.912 | 0.373 | 0.932 | 0.452 | 0.927 | 0.389 | 0.963 | 0.771 |
| Medium | SP-MIA | 0.498 | 0.050 | 0.498 | 0.051 | 0.498 | 0.049 | 0.499 | 0.051 | 0.753 | 0.460 |
| | TM-MIA | 0.857 | 0.244 | 0.749 | 0.092 | 0.861 | 0.246 | 0.757 | 0.088 | 0.976 | 0.880 |
| Heavy | SP-MIA | 0.543 | 0.061 | 0.601 | 0.059 | 0.545 | 0.078 | 0.600 | 0.067 | 0.966 | 0.846 |
| | TM-MIA | 0.782 | 0.210 | 0.740 | 0.029 | 0.792 | 0.206 | 0.741 | 0.028 | 0.931 | 0.729 |
| Extensive | SP-MIA | 0.491 | 0.047 | 0.498 | 0.000 | 0.492 | 0.049 | 0.497 | 0.043 | 0.964 | 0.963 |
| | TM-MIA | 0.687 | 0.127 | 0.542 | 0.046 | 0.688 | 0.126 | 0.542 | 0.049 | 0.862 | 0.571 |

Table 11: The results of vision models under Subset-aware Side-channel attacks and the subset selection method used here is Uncertainty (Coleman et al., 2020).

| Intensity | Setting | NN | | NN_top3 | | NN_cls | | LiRA | | CoLa | |
|---|---|---|---|---|---|---|---|---|---|---|---|
| | | AUC | TPR@5%FPR | AUC | TPR@5%FPR | AUC | TPR@5%FPR | AUC | TPR@5%FPR | AUC | TPR@5%FPR |
| Light | SP-MIA | 0.499 | 0.051 | 0.499 | 0.054 | 0.499 | 0.049 | 0.498 | 0.050 | 0.603 | 0.138 |
| | TM-MIA | 0.549 | 0.065 | 0.458 | 0.021 | 0.528 | 0.063 | 0.437 | 0.019 | 0.827 | 0.376 |
| Medium | SP-MIA | 0.494 | 0.050 | 0.498 | 0.050 | 0.494 | 0.05 | 0.496 | 0.050 | 0.811 | 0.454 |
| | TM-MIA | 0.614 | 0.073 | 0.444 | 0.020 | 0.610 | 0.072 | 0.433 | 0.019 | 0.914 | 0.703 |
| Heavy | SP-MIA | 0.554 | 0.089 | 0.625 | 0.054 | 0.560 | 0.089 | 0.627 | 0.051 | 0.959 | 0.850 |
| | TM-MIA | 0.709 | 0.13 | 0.644 | 0.025 | 0.713 | 0.130 | 0.644 | 0.025 | 0.899 | 0.644 |
| Extensive | SP-MIA | 0.501 | 0.050 | 0.506 | 0.044 | 0.502 | 0.051 | 0.502 | 0.050 | 0.929 | 0.924 |
| | TM-MIA | 0.614 | 0.117 | 0.424 | 0.000 | 0.607 | 0.118 | 0.425 | 0.03 | 0.823 | 0.575 |

