# OpenReview forum: "CoLa: A Choice Leakage Attack Framework To Expose Privacy Risks In Subset Training"
_ICLR.cc/2026/Conference — ICLR 2026 Conference Withdrawn Submission_

### Official Review · Reviewer_sTeN · 2025-10-26

**Soundness:** 3
**Presentation:** 3
**Contribution:** 3
**Rating:** 2
**Confidence:** 4

**Summary:**

This paper introduces the choice leakage attack, a unified framework for analyzing privacy leakage in subset selection. The authors propose a subset-aware side-channel attack method, which is designed for identifying both included and excluded data during data selection. Experiments show the performance of the proposed method under the selection-participation MIA setting.

**Strengths:**

1. The paper is well-presented. The authors clearly introduce the problem of the choice leakage attack, which identifies the data used for selection rather than only the training data. The distinction between TM-MIA and SP-MIA is explained in sufficient detail.

2. The experiment results are promising. The framework CoLa shows the effectiveness in detecting privacy leakage, including included data and excluded data. The authors also conduct ablation studies to analyse the performance of the method, such as window interval size and the number of clusters.

**Weaknesses:**

1. The problem formulation is impractical. The authors aim to detect both included data and excluded data used for the data selection process. However, the excluded data is not used for model training; it is still unclear why detecting non-training data is important. For example, during data selection, sample $x_1$ containing the medical information of patient A is retained for training, while sample $x_2$ containing the medical information of patient B is excluded. It is unreasonable to claim that the model leaks private information of patient B when sample $x_2$ is determined as a member. The reviewer suggests MIAs should focus on more practical settings to facilitate further advancement.
2. The method assumes access to the data selection strategy and selection rate, which is unrealistic. Given an LLM or LVM, we usually cannot reproduce the data selection procedure due to the limited transparency of technical reports. In addition, the method introduces windows to capture inclusion-stable samples. It remains unclear how the method performs when the test data contains only excluded data and no included data.
3. The description of the evaluation setup lacks clarity. Under the experimental setting, the experiment is required to split the included data (training data) and excluded data (non-training data) used for the data selection process. However, the paper seems to lack a clear description of the data splitting procedure. Especially in large language models, the training data and the pool of data selection are unpublic. It is crucial to clearly describe how to construct the included data, excluded data and outside data.

**Questions:**

1.	In the experiment of language models, how can we distinguish the included and excluded data?
2.	How do large language model sizes affect the performance of the method (e.g., 6.9B, 12B)?
3.	How do data selection methods affect the performance of the method?

---

### Official Review · Reviewer_kNfB · 2025-10-27

**Soundness:** 3
**Presentation:** 3
**Contribution:** 3
**Rating:** 4
**Confidence:** 4

**Summary:**

This paper argues that subset training (like coreset selection or data pruning) isn't "privacy-free" as many assume. The authors claim the act of choosing data leaks information, a vulnerability they call "choice leakage".  They propose two attack surfaces: TM-MIA and SP-MIA. Their new attack, CoLa, is designed to exploit this by checking if a sample is "stable"—i.e., it gets picked repeatedly across different data "windows". They show this works in both a side-channel setting (you know the selection algorithm) and a black-box one (you don't).

**Strengths:**

1. This is a genuinely new idea. Looking at the selection process itself as a privacy leak is novel.
2. The experiments are thorough—testing against nine different selection methods makes the results for the side-channel attack convincing.
3. Paper is well-written and easy to follow.

**Weaknesses:**

1. The black-box attack feels like a stretch. You assume that all selectors can be proxied by k-means clustering in an embedding space. Why?
2. This attack seems incredibly expensive. You have to re-run the selector (or clustering) on many different "windows".  For the huge datasets where pruning matters, this is totally impractical for a real attacker.
3. I'm not convinced by the LLM (SP-MIA) results. It looks like you're just distinguishing between two different datasets. This just looks like a simple distribution shift problem, not a privacy attack.
4. No defense or solutions are provided. A subsection is needed to discuss this.

**Questions:**

1. Can you prove the k-means proxy for the black-box attack is valid?
2. Isn't the CoLa attack (both side-channel and black-box) way too slow to be a practical threat on large-scale datasets?
3. You say baselines "fail" at SP-MIA, but they were built for TM-MIA (overfitting). Isn't that an unfair comparison?

---

### Official Review · Reviewer_256v · 2025-10-28

**Soundness:** 3
**Presentation:** 2
**Contribution:** 3
**Rating:** 2
**Confidence:** 2

**Summary:**

This paper challenges the common assumption that training on a data subset is inherently more private than training on a full dataset. It introduces the concept of choice leakage, a privacy risk stemming from the data selection process itself, and formally defines a new attack, Selection-Participation Membership Inference (SP-MIA), to measure it. The authors propose CoLa (Choice Leakage Attack), a multi-shot framework designed to exploit this leakage by measuring a sample's inclusion stability across various data windows.

**Strengths:**

+ Identifies a privacy risk "choice leakage" that arises from the data selection process itself
+ Introduces a new attack Selection-Participation MIA (SP-MIA)

**Weaknesses:**

- The multi-shot framework requires re-running an expensive process, such as k-means or the selection algorithm, m times to score a single sample. This multiplies the attack cost and appears impractical for large-scale datasets or adversaries testing thousands or millions of samples. Please provide a runtime and cost analysis, as well as describe any optimizations that make this approach feasible in practice.

- The black box approach treats k-means on embeddings as a generic proxy for many selection rules. There is no theoretical argument why distance to a cluster centroid should mimic criteria such as error-based, gradient-based, or decision boundary selection. Explain and justify this proxy.

- The black box score s_black(x) mixes inclusion frequency with inverse average distance to centroids. The chosen functional form is non-obvious and appears heuristic. Provide theoretical motivation, ablation studies, or comparisons with alternate scoring functions.

- The attack and especially the windowing technique imply that the adversary may need access to the full pre-selection pool D_0, including excluded samples E. State explicitly what data the adversary is assumed to have and whether access to D_0 or E is required.

- Vision results rely on a single small dataset, such as CIFAR, and on ResNet and VGG models. This does not establish generality. Evaluate on larger and more realistic datasets, such as ImageNet, and on modern architectures such as Vision Transformers.

- The distinction between Excluded E and Outside O is central for SP MIA. The paper is not perfectly explicit about how the O set was constructed or sampled. Clarify the procedure and justify why it matches realistic adversary goals.

**Questions:**

1. What are the measured runtime and computational costs of scoring m samples under the multi-shot framework?
2. What formal or empirical evidence supports using k-means on embeddings as a proxy for selection criteria like error based, gradient based, or decision boundary methods?
3. Precisely what data does the adversary hold in the threat model, and is access to the full pre selection pool D_0 or the excluded set E required?
4. How was the Outside set O constructed for SP MIA experiments and why does that sampling reflect realistic adversary objectives?

---

### Official Review · Reviewer_uiTh · 2025-11-01

**Soundness:** 2
**Presentation:** 3
**Contribution:** 2
**Rating:** 4
**Confidence:** 3

**Summary:**

This paper proposes CoLa (Choice Leakage Attack), a membership inference attack designed for models trained with subset selection. CoLa exploits the information leaked through subset choices to determine whether a sample is part of the training dataset (Training-Membership MIA) or whether it participated in the subset selection process (Selection-Participation MIA). Experiments conducted on models trained with both vision and language datasets demonstrate the effectiveness of CoLa in accurately identifying the membership of given samples.

**Strengths:**

- The study of membership inference attacks in subset training is an interesting and important problem. It raises concerns that even when the training data is reduced, the risk of privacy leakage may still remain high.

- The paper is clearly written and easy to follow.

**Weaknesses:**

- The paper primarily demonstrates the effectiveness of the proposed method through experimental results. However, it lacks analysis or explaining why CoLa achieves such high performance in membership inference attacks.

- The experiments are conducted on relatively small and outdated datasets, such as CIFAR-10, which may limit the generality of the conclusions regarding the effectiveness of the proposed method. In addition, the baseline methods used for comparison are quite outdated.

**Questions:**

I would consider raising my rating if the authors are able to address my concerns

Q1. Table 1 presents the results of different methods under various selection ratios for subset training. I am curious how these results compare to full training, i.e., when the model is trained on the entire dataset without subset selection. Would subset training help reduce privacy leakage compared to full training?

Q2. The table also shows an unusual trend: for most existing methods, the performance of TM is typically higher than or roughly comparable to SP (within about 1%). However, for the proposed method CoLa, when r = 0.6 and r = 0.8, SP achieves significantly higher attack performance than TM. Could the authors provide an explanation for this result?

Q3. In the vision domain, experiments are only conducted on CIFAR-10, and the baseline methods used are quite outdated (i.e., MIAs published before 2022). Could the authors include results on larger and more recent datasets, as well as comparisons with newer membership inference attacks?

Q4. In the black-box setting, it is unclear how the embedding model is trained. Does it use the same dataset D_0  as the target model, or is it trained on a public dataset? How does the choice of training dataset (D_0/ D_0 + non-member data/ non-member data/ public dataset) affect the results?

Q5. Could the authors provide analysis or explaining why CoLa achieves such high performance in membership inference attacks.

Q6. I am interested in how CoLa performs when evaluated on models that incorporate membership inference defense methods?

---

### Note · Authors · 2026-01-02

I have read and agree with the venue's withdrawal policy on behalf of myself and my co-authors.